# Yogurt Alleviates Cyclophosphamide-Induced Immunosuppression in Mice through D-Lactate

**DOI:** 10.3390/nu16091395

**Published:** 2024-05-06

**Authors:** Xinru Du, Yongheng Yan, Yufeng Dai, Ruijie Xu

**Affiliations:** 1State Key Laboratory of Food Science and Resources, Jiangnan University, Wuxi 214122, China; 7200112055@stu.jiangnan.edu.cn; 2School of Food Science and Technology, Jiangnan University, Wuxi 214122, China; 3School of Public Health, Shandong First Medical University, Jinan 271016, China; 4Global Health Institute, School of Public Health, Xi’an Jiaotong University, Xi’an 710061, China; xurj0087@xjtu.edu.cn

**Keywords:** yogurt, D-lactate, cyclophosphamide (CTX), immunosuppression, gut microbiota

## Abstract

Numerous studies have investigated the immunomodulatory effects of yogurt, but the underlying mechanism remained elusive. This study aimed to elucidate the alleviating properties of yogurt on immunosuppression and proposed the underlying mechanism was related to the metabolite D-lactate. In the healthy mice, we validated the safety of daily yogurt consumption (600 μL) or D-lactate (300 mg/kg). In immunosuppressed mice induced by cyclophosphamide (CTX), we evaluated the immune regulation of yogurt and D-lactate. The result showed that yogurt restored body weight, boosted immune organ index, repaired splenic tissue, recovered the severity of delayed-type hypersensitivity reactions and increased serum cytokines (IgA, IgG, IL-6, IFN-γ). Additionally, yogurt enhanced intestinal immune function by restoring the intestinal barrier and upregulating the abundance of Bifidobacterium and Lactobacillus. Further studies showed that D-lactate alleviated immunosuppression in mice mainly by promoting cellular immunity. D-lactate recovered body weight and organ development, elevated serum cytokines (IgA, IgG, IL-6, IFN-γ), enhanced splenic lymphocyte proliferation and increased the mRNA level of T-bet in splenic lymphocyte to bolster Th1 differentiation. Finally, CTX is a chemotherapeutic drug, thus, the application of yogurt and D-lactate in the tumor-bearing mouse model was initially explored. The results showed that both yogurt (600 μL) and D-lactate (300 mg/kg) reduced cyclophosphamide-induced immunosuppression without promoting tumor growth. Overall, this study evaluated the safety, immune efficacy and applicability of yogurt and D-lactate in regulating immunosuppression. It emphasized the potential of yogurt as a functional food for immune regulation, with D-lactate playing a crucial role in its immunomodulatory effects.

## 1. Introduction

Cyclophosphamide (CTX) is a broad-spectrum anticancer drug. It exerts the effects by interfering with the DNA synthesis of tumor cells, thereby inhibiting their proliferation and survival. However, while killing tumor cells, CTX also exhibits toxic side effects of immunosuppression [1,2]. CTX suppresses the immune system through diverse mechanisms, including encompassing the inhibition of immune cell functionality, attenuation of immune responses in T cells and B cells, as well as modulation of immune globulins, chemokines, and cytokines [3]. Immunosuppression is correlated with various chronic diseases, such as infections and cancers [4]. Furthermore, an imbalance of immune regulation results in gastrointestinal inflammation and intestinal flora disorders [5,6]. Therefore, it is imperative to identify effective strategies for preventing and alleviating immunosuppression.

Yogurt is a typical functional food. Epidemiological studies have revealed that yogurt can decrease the incidence of type 2 diabetes, metabolic syndrome and heart disease [7,8,9,10]. Previous researches on the immunoregulation of yogurt have focused on bioactive nutrients, such as fat acids, phytosterols, proteins, vitamins and probiotics [7,11,12]. However, D-lactate has been almost ignored as a byproduct of bacteria fermentation in yogurt. D-lactate is a configuration of chiral lactate. D-lactate typically has two origins in the human body. A portion is the dietary intake digested by bacteria in conventional foods, such as yogurt or pickles [13,14]. Lactobacillus bulgaricus can convert 90% of pyruvate into D-lactate [14] during yogurt manufacturing. Endogenous D-lactate is processed by closely related gut bacteria [15]. According to prevailing research, it is widely accepted that the human body exclusively harbors L-lactate dehydrogenase, which possesses rapid metabolic capacity towards L-lactate, while lacking D-lactate dehydrogenase. Consequently, the elevated concentration of D-lactate in plasma results in the acidosis. However, our recent research has demonstrated the daily intake of 2000 mg/kg of D-lactate does not affect the normal growth of mice. Furthermore, in comparison to L-lactate, D-lactate exhibits stronger anti-inflammatory properties due to its pharmacokinetic advantages [16]. Additionally, the recent study also reports that D-lactate plays a crucial role in preventing colitis-associated colon cancer in yogurt [17]. Hence, D-lactate is plausibly a potential dietary nutrient factor.

This research studies the actions of yogurt on immunosuppression by constructing three animal models (healthy mice, immunosuppressive mice and tumor-bearing mice), through D-lactate-dependent mechanisms. This study provides novel insights into alleviating immunosuppression and supports the benefits of traditional fermented foods and the development of D-lactate.

## 2. Materials and Methods

### 2.1. Chemicals and Reagents

Cyclophosphamide and Levamisole hydrochloride (LM) were sourced from Aladdin. D-sodium lactate, ConA and LPS were purchased from Sigma-Aldrich Co., St. Louis, MO, USA. Kits for measuring creatinine (CRE), urea nitrogen (BUN), aspartate aminotransferase (AST), and alanine aminotransferase (ALT) were acquired from the Nanjing Jiancheng Institute of Biological Engineering (Nanjing, China). Sheep red blood cells (SRBC) were obtained from Nanjing Senbeijia Biological Co., Ltd. (Nanjing, China). Spleen lymphocyte extraction kit was purchased from Beijing Solarbio Science & Technology Co., Ltd. (Beijing, China). CCK-8 Cell Counting Kit was purchased from Nanjing Vazyme Biotech Co., Ltd. (Nanjing, China). IgA, IgG, IL-4, IL-6, and IFN-γ Elisa kits were purchased from R&D systems. The CT26 cell line was derived from the American Type Culture Collection (ATCC) (maintained in a humidified incubator at 37 °C and 5% CO_2_).

### 2.2. Yogurt Preparation

A simulated commercial yogurt was prepared according to a previous study [18,19]. Lactobacillus bulgaricus and Streptococcus thermophilus bacterial strains were mixed in a 1:1 ratio and incubated at 39 °C for 18 h in 10% skim milk, followed by a 24h incubation at 4 °C. At the end of this process, the total viable count was 9.16 lg (CFU/mL) consistent with the former research [20].

### 2.3. Determination of D-Lactate in Yogurt by High-Performance Liquid Chromatography (HPLC)

The sample pretreatment was drawing upon established approache with moderate modification [21]. Samples were centrifuged at 12,000 r/min for 15 min. Subsequently, 10% trichloroacetic acid was added to the samples and allowed to react for two hours before being centrifuged at 12,000 r/min for 20 min. Following this, the samples were filtered using a 0.22 μm filtration membrane and degassed through ultrasonic means before analysis with HPLC. The separation of L-lactate and D-lactate was achieved by Chirex 3126 (D) penicillamine LC Column [21,22].

### 2.4. Animal Experiment

Female SPF BALB/c mice weighing approximately 20.0 ± 2.0 g and aged 7 weeks were procured from Beijing Vital River Laboratory Animal Technology Co., Ltd. (Beijing, China). The mice were housed in cage and a controlled environment with a temperature of 22 ± 2 °C, a humidity of 50 ± 10%, and a 12-light-dark cycle, following the Guidelines for Keeping Experimental Animals issued by the Chinese government. The Experimental Animal Welfare and Ethics Committee of Jiangnan University approved the study (license numbers JN.No20220315c0900425(012), JN.No20220615c0801001(198) and JN.No20221130b07002209467).

#### 2.4.1. Animal Experiment I

After seven days of adaptive feeding, total of 42 mice aged 5-week-old were randomly assigned to 6 groups, with 7 mice per group. Daily records were maintained for the baseline condition and body weight of the mice for 22 days. Based on animal welfare principles, at the end of the experiment, the liver, spleen, and thymus were extracted and weighed for calculation. The toxicological evaluation of yogurt and D-lactate was conducted by assessing hepatic function (represented by ALT and AST) and renal function (represented by BUN and CRE). CRE, BUN, AST, and ALT levels were measured using appropriate kits and following the manufacturer’s instructions.

#### 2.4.2. Animal Experiment II

After seven days of adaptive feeding, total of 108 mice aged 7-week-old mice were randomly assigned to 9 groups, with 12 mice per group. In each treatment group, immunosuppression was induced by intraperitoneal injection of CTX (80 mg/kg) for 3 days. The successful establishment of an immunosuppression model was recognized when the body weight of the mice decreased by 15% of the original. Different doses of yogurt and D-lactate were supplied during the following days. Based on animal welfare principles, on day 17 of the experiment, 6 mice were randomly selected to inject intraperitoneally with 0.2 mL of 5% (*v*/*v*) sheep red blood cells (SRBC). After four days, serum was collected from the eye orbit for the HC_50_ value assessment. Simultaneously, 20 μL of 20% (*v*/*v*) SRBC was injected into the right hindfoot pad for the measurement of the delayed-type hypersensitivity (DTH) reaction. Serum, spleen, thymus, and fecal samples were collected for biochemical analysis at the end of the experiment.

#### 2.4.3. Animal Experiment III

After seven days of adaptive feeding, total of 70 mice aged 7-week-old mice were randomly assigned to 7 groups, with 10 mice per group. For the tumor-bearing model, after adjusting the CT26 cell at the logarithmic growth phase suspension to a concentration of 1 × 10^6^ cells/mL with PBS, the cell suspension was inoculated subcutaneously into the right axilla of mice, with a dose of 0.2 mL per mouse. The mice were then bred for seven days, and those who developed a tumor volume of about 50 ± 16 mm^3^ were deemed to have successfully established models [23]. Subsequently, the mice were injected with CTX every three days and orally administered yogurt and D-lactate daily. Daily observations were made of the behavior, body weight, and tumor size of all mice. At the end of the test, the tumor volume did not exceed the ethical limits. Based on animal welfare principles, serum, spleen, thymus, and tumor tissues were collected for biochemical analysis at the end of the experiment.

### 2.5. Methods for Physiological and Pathological Analysis

#### 2.5.1. ELISA Assay

The IL-4, IL-6, IFN-γ, IgA, and IgG cytokines were measured using a commercial ELISA kit and following the manufacturer’s protocols.

#### 2.5.2. Histopathological Staining

The sample tissues were fixed overnight at room temperature with 10% formaldehyde, washed with distilled water, subjected to dehydration using a gradient of alcohol, and embedded in paraffin wax. Paraffin-embedded slices were stained. Images were taken using a light microscope.

#### 2.5.3. RT-qPCR Analysis 

Total RNA was extracted using a Trizol reagent and assessed for purity and integrity through NanoDrop and gel electrophoresis. RNA was converted to cDNA using a reverse transcription kit. Gene expression levels were evaluated using a real-time quantitative PCR system. Gene-specific primers were listed in Table 1. The relative quantification of the target gene was determined by comparing it to β-actin and calculating the results using the 2^−∆∆Ct^ method [24].

#### 2.5.4. Proliferation Assay of Splenic Lymphocytes 

Mouse spleens were aseptically removed, and lymphocytes were collected using the Animal Splenic Lymphocyte Isolation Kit from Beijing Solarbio Science & Technology Co., Ltd. (Beijing, China). [25]. Cells were incubated in 96-well plates with ConA (10 μg/mL) and LPS (5 μg/mL) for 24 h, followed by treatment with CCK-8 solution and another 2 h incubation [26]. The absorbance measurement at 450 nm was quantified using a Thermo Scientific Varioskan Flash spectrophotometer (Waltham, Massachusetts, America).

#### 2.5.5. Delayed-Type Hypersensitivity (DTH)

On day 17 of the experiment, 0.2 mL of 5% (*v*/*v*) SRBC was injected intraperitoneally into the mice, and the thickness of the right hindfoot pad was measured utilizing vernier calipers. Four days after the initial injection, 20 μL of 20% (*v*/*v*) SRBC was administered to the right hindfoot pad. The resulting increase in footpad thickness was determined 24 h after the second injection and utilized as an indicator of DTH.

#### 2.5.6. Serum Hemolysin

Mice were received an injection of 0.2 mL of 5% (*v*/*v*) SRBC, and after four days, serum was obtained from blood collected from their eye orbit. The serum was diluted with SA buffer (1:5) and added to a 96-well culture plate with a control buffer. Then, SRBC and guinea pig serum were added to each well, and the supernatant was mixed with Drabkin’s solution. After resting for 10 min, the optical density of each well was measured at 540 nm using an automatic microplate reader to evaluate serum hemolysin levels.

#### 2.5.7. Gut Microbiota Analysis

DNA was extracted from the cecal contents using the FastDNA™ Spin Kit for Feces. Agarose electrophoresis was used to detect gDNA integrity and nucleic acid concentration. PCR was performed using universal primer pairs on the V3–V4 high variant region of the bacterial 16S rRNA gene [27]. The 16SrDNA sequence data were processed using QIME V19.1 to remove any raw sequences that did not meet specific criteria. High-quality reads were selected for bioinformatics analysis, and operational taxonomic units (OTUs) were formed by clustering all valid reads with similarities greater than 97%. OTUs were classified according to the Greengenes database, and the R package was utilized for α-diversity, β-diversity, and species screening based on the abundance of OTUs.

### 2.6. Statistical Analyses

Statistical analyses were performed using GraphPad Prism 8.0. A one-way analysis of variance (ANOVA), followed by the Bonferroni procedure, was applied to compare means for multiple group comparisons. At the same time, a Student’s *t*-test was used to compare two independent groups. The data were expressed as mean ± SEM, with a significance level set at *p* ≤ 0.05.

## 3. Results

### 3.1. Detection of D-Lactate in Yogurt

The findings indicated a total lactate content of 15.43 mg/mL in yogurt, with the specific D-lactate content measured at 5.90 mg/mL. After conversion, the proportion of total lactate in yogurt was 1.50%, with the proportion of D-lactate being 0.57%. Additional details concerning the dosage of yogurt and D-lactate were found in the Appendix A.

### 3.2. Animal Experiment I: Toxicity Evaluation of Yogurt and D-Lactate in Healthy Mice

In Table 2, mice treated with yogurt and D-lactate maintained stable body weights within the normal range. Yogurt (600 μL) and D-lactate (300 mg/kg) increased immune organ indexes of the spleen and thymus without causing any toxic effects on the hepatic (represented by ALT and AST) and renal function (represented by BUN and CRE). Comparable results were observed in the Positive Control group treated with LM, a recognized immunoregulatory compound [28]. The findings showed the oral safety of yogurt (600 μL) and D-lacate (300 mg/kg) in healthy mice.

### 3.3. Animal Experiment II: Evaluation of Immune Efficacy of Yogurt and D-Lactate in Immunosuppressive Mice Induced by CTX

#### 3.3.1. Yogurt Supplementation Ameliorated CTX-Induced Immunosuppression in Mice

The experimental design was displayed (Figure 1A). Yogurt dose-dependently recovered the body weight (Figure 1B) and facilitated the growth of spleen and thymus compared to the MC group (Figure 1C). The spleen of the MC group exhibited the reduced number of splenic corpuscles and lymphocytes, dispersed germinal centers, and the unclear distinction between the red and white pulp. However, this trend was reversed upon yogurt administration (Figure 1D). Besides these analyses, the yogurt administration resulted in a dose-dependent increase in serum IgA, IgG, IFN-γ, and IL-6 levels (Figure 1E–H). The DTH reaction exhibited similar alterations. Treatment with medium or high dosages of yogurt resulted in a significant increase in footpad thickness compared to the MC group (Figure 1I). 

#### 3.3.2. Yogurt Supplementation Regulated Intestinal Immunity and Gut Microbiota in CTX-Induced Mice

As was showed in Figure 2, the standard group showed typical glandular structures and slender villi arranged tightly and completely in the intestines. In contrast, the MC group suffered from significant injury to the intestinal wall, featuring shortened and detached villi. However, the medium and high dosage of yogurt groups exhibited restored villi length (Figure 2A). Yogurt upregulated the expression of MUC-2 (Figure 2B) and claudin5 (Figure 2C), enhancing the integrity of the intestinal barrier. The detected biomarkers of the ratio of IL-4 and IFN-γ related to immune responses in the intestine were significantly improved by yogurt (Figure 2D). Data from the α-diversity (Figure 2E) of Chao1, Shannon, Simpson and ACE index indicated that yogurt modulated the overall diversity of gut microbiota. The β-diversity analysis showed that the gut microbiota composition in yogurt group was distinct from the NC and MC groups (Figure 2F), characterized by 4187 unique OTUs (Figure 2G). At the phylum level, the ratio of Firmicutes to Bacteroidetes (F/B) was a biomarker to assess pathological status. F/B tended to increase moderately by yogurt supplementation (Figure 2H). At the genus level, according to the heat map analysis (Figure 2I), yogurt consumption resulted in a dominate abundance of beneficial bacteria of Lactobacillus, Streptococcus, Bifidobacterium, Akkermansia and Ruminococcus. Yogurt especially increased the abundance of Bifidobacterium and Lactobacillaceae (Figure 2I,J). In addition, it mitigated the harmful bacteria such as Desulfovibrio, Staphylococcus, and Escherichia. We examined the D-lactate level by HPLC in the gut contents to assess the potential involvement of D-lactate in intestinal immunity. The results showed high D-lactate levels in the feces of yogurt groups but minimal changes in D-lactate groups (Figure 2K).

#### 3.3.3. D-Lactate Supplementation Ameliorated Immunosuppression in CTX-Induced Mice

The experimental design was displayed (Figure 3A). The mice administered with D-lactate showed a significant increase in body weight (Figure 3B) and organ development (Figure 3C) dose-dependently. D-lactate ameliorated splenic tissue damage induced by CTX (Figure 3D). Furthermore, splenic lymphocytes were isolated and stimulated by ConA and LPS in vitro to assess transformation capacity of T cells and B cells respectively. The splenic lymphocyte proliferation was measured by SI values. The result indicated that treatment with D-lactate significantly boosted the ConA-stimulated proliferation responses of T cells in immunosuppressive mice (Figure 3E). No similar results were observed in LPS-stimulated proliferation responses of B cells. The results demonstrated that CTX caused a decrease in the relative mRNA expression levels of T-bet, GATA3, Foxp3, and RORγt in splenic lymphocytes mice. The medium and high dosage of D-lactate increased T-bet mRNA level (Figure 3F), while only the high dosage of D-lactate increased GATA3 mRNA level in splenic lymphocytes (Figure 3G). However, no alteration was observed in the expression level of Foxp3 (Figure 3H) and RORγt (Figure 3I). CTX inhibited immune cell cytokine production and resulting in immunosuppression. However, the suppression of cytokine levels (Figure 3J–M) was significantly eased by D-lactate. The HC_50_ values were used to assess murine humoral immune function, as reflected by the serum hemolysin index [29]. The levels of HC_50_ were not differentially affected by the three D-lactate treatments (Figure 3N). 

### 3.4. Animal Experiment III: Evaluation of Immune Efficacy of Yogurt and D-Lactate in CT26 Tumor-Bearing Mice 

The experimental design was displayed (Figure 4A). Yogurt effectively ameliorated CTX-induced body weight loss in CT26 tumor-bearing mice (Figure 4B). Yogurt restored the organ development (Figure 4C) and promoted the cytokine secretion of IL-6 and IFN-γ (Figure 4D) in tumor-bearing mice treated with CTX. The tumor tissue (Figure 4E) and the analysis of tumor weight on the day 20 (Figure 4F) revealed that the tumor inhibition rates of the CTX and C + Y groups were 68.02 ± 5.6% and 71.34 ± 4.8%, respectively. The Bax/Bcl-2 ratio expression of colorectal cancer was enhanced with the statistical difference in the CTX and C + Y groups compared to the control group (Figure 4G). This observation suggests that yogurt consumption did not compromise the antitumor efficacy of CTX. Similarly, the experimental design of D-lactate was displayed (Figure 4H). Significant variations in body weight were observed among the groups (Figure 4I). The C + D group enhanced organ index (Figure 4J) and cytokine secretion levels (Figure 4K) compared to the CTX group. D-lactate did not impact the antitumor effectiveness of CTX, as evidenced by tumor growth inhibition rates of 70.28% ± 2.6% for the C + D group (Figure 4L,M) and the promotion of tumor apoptosis (Figure 4N). In summary, yogurt and D-lactate alleviated the immunosuppressive effect of CTX in tumor-bearing mice and did not affect the antitumor efficacy of CTX. 

## 4. Discussion

The present study provided direct evidence of yogurt to alleviate the CTX-induced immunosuppression. Furthermore, D-lactate was identified as a potential substance responsible for the immunomodulatory effects of yogurt. Although the results appeared promising, some statements and discussion still required to be supplied.

The first question was regard to the dosage information of yogurt and D-lactate. Appendix A displayed the lactate concentration in simulated commercial yogurt. The data illustrated that the total lactate content in yogurt was 1.50% and D-lactate content was 0.57%. According to FDA, the recommended daily intake of yogurt for a human weighed 80 kg was 150–200 g, which was converted to 300–400 μL per day in mice. Accordingly, a human weighing 80 kg consuming 200 g yogurt a day was equivalent to a mouse weighing 20 g ingested with about 4 mg D-lactate per day. This served as the basis for the dosage of yogurt and D-lactate in the experimental design.

The second question revolved the modulation of gut microbiota in immunosuppressive mice by yogurt. From the data of α-diversity, the increase of Chaos, Simpson and ACE indexes in the model group mice were possibly attributed to the upregulated abundance of pathogenic microbiota [30] (Appendix A). It was speculated that the increased indexes of Chaos, Shannon, Simpson and ACE in yogurt group might contribute to the enhanced diversity of beneficial gut bacteria [7,31]. Further β-diversity analysis of the gut microbiota has validated that hypothesis (Appendix A). The findings of β-diversity coincided precisely with the α-diversity results. Particularly, Lactobacillus and Bifidobacterium showed a notable increase, which are potentially considered to hydrolyze carbohydrates in the intestine to produce D-lactate [32]. Correlational probiotic strains, including Lactobacillus reuteri ATCC 55730 [33], Lactobacillus johnsonii La [34], and *L. reuteri* DSM 17938 [35,36] have been reported to produce D-lactate. The results of D-lactate detection in intestinal contents provided further inference. Therefore, it was reasonable to speculate that the increasing D-lactate concentration in feces was potentially associated with the remodeling of the intestinal flora. Furthermore, studies have demonstrated that microbial D-lactate can affect macrophage function locally and maintain mucosal integrity within the gut [37]. Thus, yogurt consumption enhanced the intestinal immune function by hypothetically reshaping the gut microbiota to boost beneficial probiotics producing D-lactate in the intestine. Nevertheless, it cannot be ruled out that the gut immunity might be modulated by the oral administration of D-lactate through other pathways.

The third question was to discuss the mechanism by which D-lactate alleviated immunosuppression. Immunocytes can be categorized into T cells and B cells. Our findings demonstrated that D-lactate enhanced the immune response of splenic T cells stimulated by ConA, thereby augmenting their capacity for lymphocyte transformation. However, no significant differences were observed regarding the immune response of splenic B cells. Consequently, we further examined the expression levels of nuclear transcription factors of the T cell family in the mouse spleen. T-bet was a nuclear transcription factor specifically expressed in Th1 cells, regulating cellular immune function. GATA3 was a key transcription factor involved in the development of Th2 cells, regulating humoral immunity. A delicate balance between Th1 and Th2 cells was crucial for the healthy host [38,39]. Foxp3 was a transcriptional regulatory factor that was critical for the development and suppressive function of regulatory T cells. RORγt was a lineage-specific transcription factor for T helper 17 cells. The results demonstrated that the ingestion of medium and high doses of D-lactate resulted in an increase in the expression level of T-bet, while only a high dose of D-lactate elevated the expression level of GATA3. This indicated that D-lactate regulated the differentiation of Th cells towards Th1 cells and alleviated immunosuppressive by enhancing cellular immunity. These findings were consistent with the negative results of HC_50_ (an indicator for measuring humoral immunity) and positive results of DTH (the vivo detection of cellular immunity) [40]. Therefore, D-lactate exerted its mitigating effect on immunosuppression by promoting Th cell differentiation to Th1 and enhancing the cellular immunity of splenic lymphocytes. 

The fourth question addressed why the tumor model was constructed. In healthy mice and immunosuppression mice, we verified the safety and immune efficacy of daily intake of 600 μL yogurt or 300 mg/kg D-lacate in mice. However, whether yogurt or D-lactate could be applied to clinical cancer chemotherapy patients remains debatable. In the tumor model, we conducted preliminary exploration of the therapeutic effects of yogurt and D-lactate on CTX chemotherapy. Our research findings demonstrated that yogurt and D-lactate can alleviate CTX-induced immunosuppression to a certain extent in the mouse tumor model, and they had no detrimental effects on the antitumor efficacy of CTX. The findings from this study have provided preliminary data support for the clinical application of yogurt and D-lactate in alleviating immunosuppressive.

Although our research has successfully assessed the mitigating effects of yogurt and D-lactate on immunosuppression, several limitations remain. The immune efficacy of other bioactive ingredients in yogurt cannot be denied, but this study focuses on evaluating the benefits of D-lactate in yogurt, aiming to develop D-lactate as a new type of food functional factor. The potential benefits of other components in yogurt should be explored in the future. Additionally, further exploration of the specific production of D-lactate by gut microbiota was valuable for developing D-lactate into a new type of food nutrient factor. Furthermore, mechanistic investigations at the molecular level have elucidated that lactate exerted its effects through specific binding to its receptor GPR81 [41]. Using flow cytometry to subtype lymphocytes might be advantageous for investigating the involvement of GPR81 in immune responses at a more advanced scientific level. On the other hand, despite our research findings has demonstrated the positive adjunctive effect of D-lactate on chemotherapy, clinical trials were warranted to validate these findings and establish the utility of yogurt as an immunomodulatory food product. D-lactate remained to determine whether it was comparable to the clinical drug Mesna.

## 5. Conclusions

In summary, this study effectively demonstrated the mitigating effects of yogurt on immunosuppression. The alleviation of yogurt on immunosuppression is partially attributed to the enhanced cellular immunity of D-lactate on splenic lymphocytes. This article provides a new interpretation of yogurt as a functional food exerting physiological effects, making a positive contribution to the functional food segment.

## Figures and Tables

**Figure 1 nutrients-16-01395-f001:**
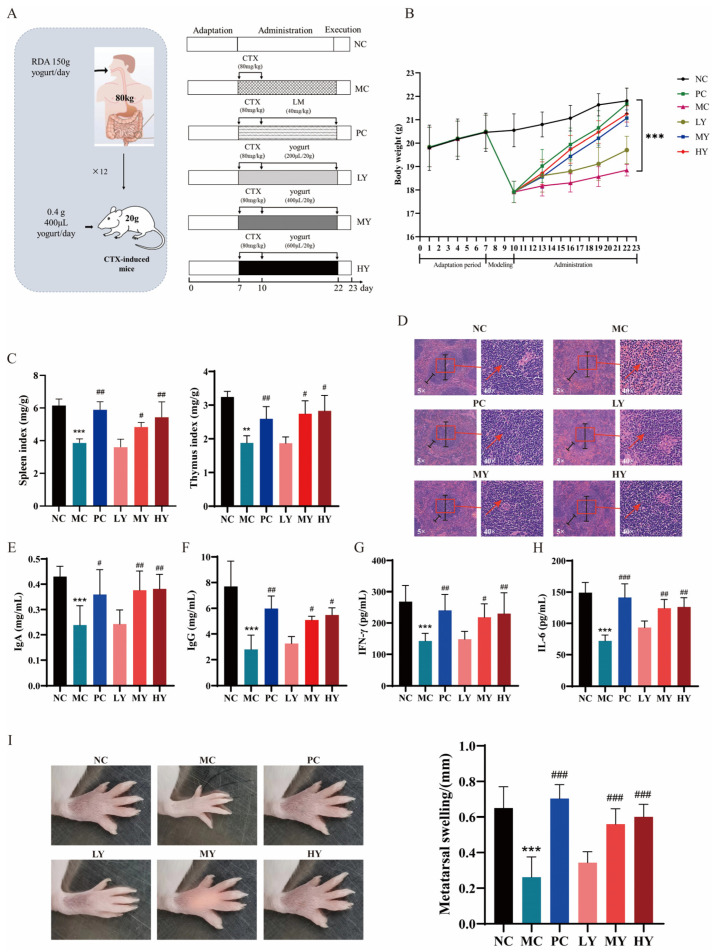
Yogurt supplementation ameliorated CTX-induced immunosuppression in mice. (**A**) Experimental design. NC: the normal control group; MC: the model control group (cyclophosphamide, 80 mg/kg bw); PC: the positive control group (levamisole hydrochloride, 40 mg/kg bw); LY: Low-dose of yogurt group (200 μL); MY: Medium-dose of yogurt group (400 μL); HY: High-dose of yogurt group (600 μL). (**B**) Body weight. (**C**) Spleen and thymus indexes. (**D**) Histopathology observation of the spleen (original magnification: ×5 and ×40). (**E**–**H**) Serum cytokines of IgA, IgG, IL-6 and IFN-γ. (**I**) DTH reaction and Metatarsal swelling, ** *p* < 0.01 and *** *p* < 0.005 vs. NC group. ^#^
*p* < 0.05, ^##^
*p* < 0.01 and ^###^
*p* < 0.005 vs. MC group.

**Figure 2 nutrients-16-01395-f002:**
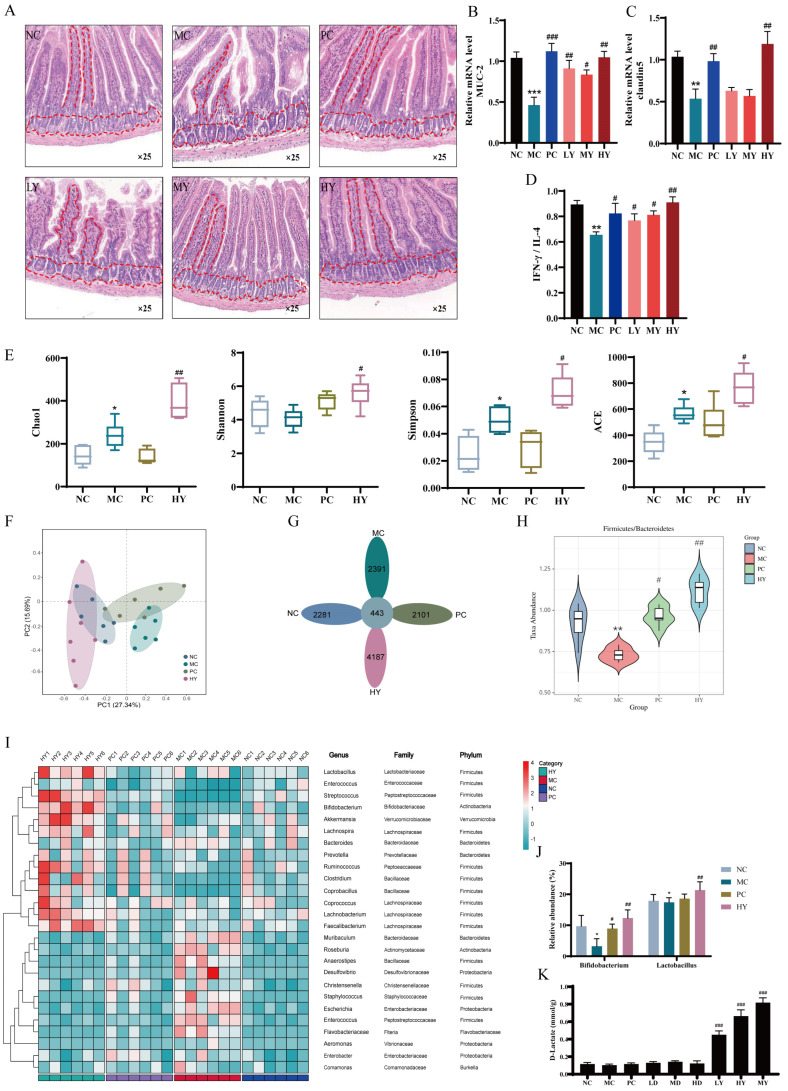
Yogurt supplementation regulated intestinal immunity and gut microbiota in CTX-induced mice. NC: the normal control group; MC: the model control group (cyclophosphamide, 80 mg/kg bw); PC: the positive control group (levamisole hydrochloride, 40 mg/kg bw); LY: Low-dose of yogurt group (200 μL); MY: Medium-dose of yogurt group (400 μL); HY: High-dose of yogurt group (600 μL); LD: Low-dose of D-lactate group (75 mg/kg bw); MD: Medium-dose of D-lactate group (150 mg/kg bw); HD: High-dose of D-lactate group (300 mg/kg bw). (**A**) Histopathology observation of the ileum tissue (original magnification: ×25). (**B**) Relative mRNA level of MUC-2. (**C**) Relative mRNA level of claudin5. (**D**) Cytokines of IFN-γ/IL-4 in ileum tissue. (**E**) Alpha diversity of Chao 1, Shannon, Simpson and ACE indexes. (**F**,**G**) PCA and Venn diagram of gut microbiota. (**H**) The ratio of Firmicutes to Bacteroidetes abundance. (**I**) Heatmap. (**J**) The relative abundance of Bifidobacterium and Lactobacillus. (**K**) The D-lactate concentration in feces. * *p* < 0.05, ** *p* < 0.01 and *** *p* < 0.005 vs. NC group. ^#^
*p* < 0.05, ^##^
*p* < 0.01 and ^###^
*p* < 0.005 vs. MC group.

**Figure 3 nutrients-16-01395-f003:**
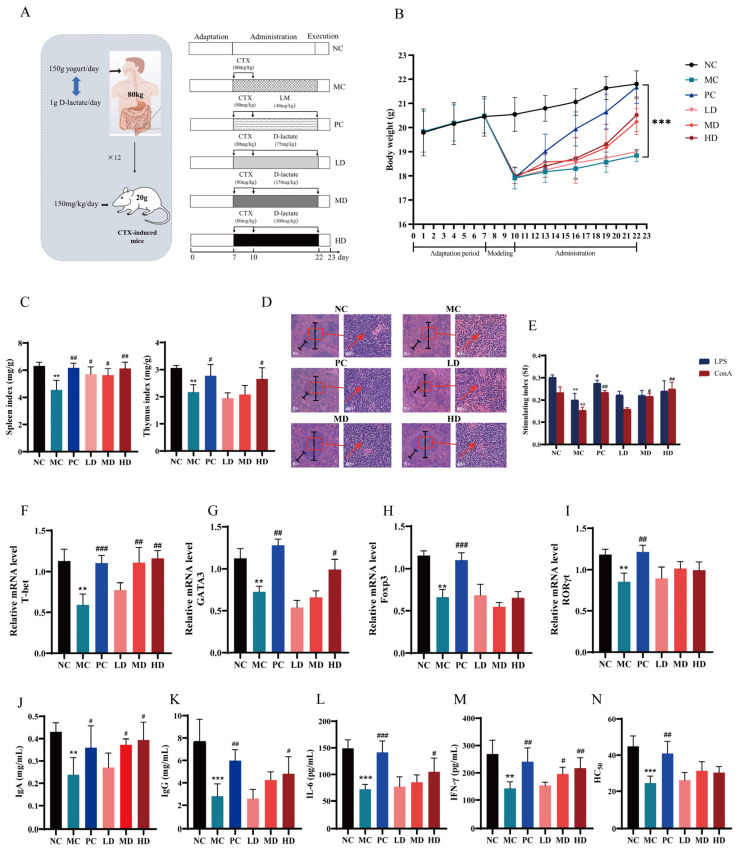
D-lactate supplementation ameliorated immunosuppression in CTX-induced mice. (**A**) Experimental design. NC: the normal control group; MC: the model control group (cyclophosphamide, 80 mg/kg bw); PC: the positive control group (levamisole hydrochloride, 40 mg/kg bw); LD: Low-dose of D-lactate group (75 mg/kg bw); MD: Medium-dose of D-lactate group (150 mg/kg bw); HD: High-dose of D-lactate group (300 mg/kg bw). (**B**) Body weight. (**C**) Spleen and thymus indexes. (**D**) Histopathology observation of the spleen (original magnification: ×5 and ×40). (**E**) Stimulating index (SI). (**F**–**I**) Relative mRNA level of T-bet, GATA3, Foxp3 and RORγt. (**J**–**M**) Serum cytokines of IgA, IgG, IL-6 and IFN-γ. (**N**) Serum half hemolysis value (HC_50_), ** *p* < 0.01 and *** *p* < 0.005 vs. NC group. ^#^
*p* < 0.05, ^##^
*p* < 0.01 and ^###^
*p* < 0.005 vs. MC group.

**Figure 4 nutrients-16-01395-f004:**
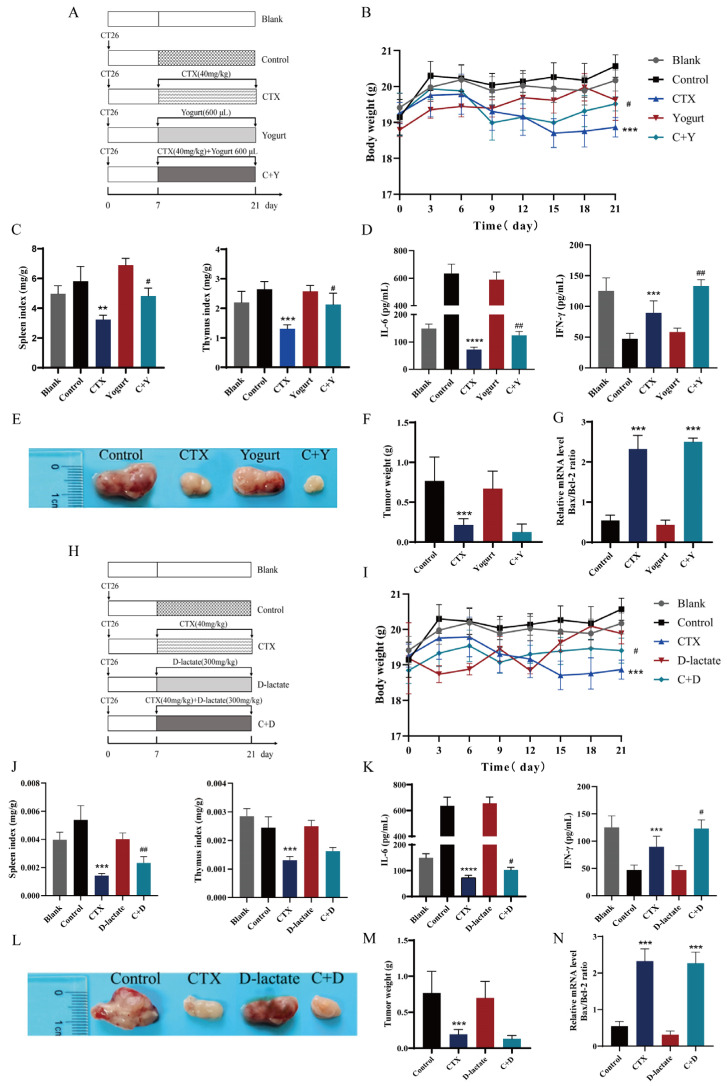
Evaluation of immune efficacy of yogurt and D-lactate in CT26 tumor-bearing mice. (**A**,**H**) Experimental design, Blank: healthy mice; Control: tumor model group; CTX: cyclophosphamide (40 mg/kg bw); C + Y: cyclophosphamide + yogurt (cyclophosphamide, 40 mg/kg bw; yogurt, 600 μL); C + D: cyclophosphamide + D-lactate (cyclophosphamide, 40 mg/kg bw; D-lactate, 300 mg/kg bw). (**B**,**I**) Body weight. (**C**,**J**) Immune Organ index. (**D**,**K**) Serum cytokines of IL6 and IFN-γ. (**E**,**L**) Images of tumor tissues at treatment termination. (**F**,**M**) tumor weight at 20 d. (**G**,**N**) Relative mRNA level of Bax/Bcl-2 ratio, ** *p* < 0.01, *** *p* < 0.001 vs. control group. ^#^
*p* < 0.05, ^##^
*p* < 0.01 vs. CTX group. **** *p* < 0.0001.

**Table 1 nutrients-16-01395-t001:** Primer sequences.

Genes	Primer Forward	Primer Reverse
MUC-2	GATTCGAAGTGAAGAGCAAG	CACTTGGAGGAATAAACTGG
Claudin5	GAGAGGAACTACCCTTATGCC	ATTGAGTAATTAAACGGGACAGG
T-bet	CGTTTCTACCCCGACCTTCC	ATGCTCACAGCTCGGAACTC
GATA-3	AAGCTCAGTATCCGCTGACG	GATACCTCTGCACCGTAGCC
Bax	ACAGATCATGAAGACAGGGG	CAAAGTAGAAGAGGGCAACC
Bcl-2	ATGTGTGTGGAGAGCGTCAAC	AGACAGCCAGGAGAAATCAAAC
β-actin	TGCTCTCCCTCACGCCATC	GAGGAAGAGGATGCGGCAGT

**Table 2 nutrients-16-01395-t002:** Evaluation of immune efficacy and toxicity of yogurt and D-lactate in healthy mice.

	Normal Control	Positive Control	Yogurt (200 μL)	Yogurt (600 μL)	D-Lactate (75 mg/kg)	D-Lactate (300 mg/kg)
Body weight (g)	28.01 ± 0.8	29.11 ± 1.2	28.31 ± 0.7	29.21 ± 0.5	27.26 ± 1.9	28.09 ± 1.1
Thymus index (%)	0.27 ± 0.06	0.36 ± 0.09 *	0.28 ± 0.12	0.32 ± 0.03 *	0.28 ± 0.04	0.38 ± 0.02 *
Spleen index (%)	0.32 ± 0.05	0.42 ± 0.12 *	0.38 ± 0.17 *	0.40 ± 0.19 *	0.32 ± 0.09	0.40 ± 0.10 *
Liver index (%)	4.31 ± 0.21	4.51 ± 0.09	4.33 ± 0.19	4.68 ± 0.16	4.53 ± 0.14	4.61 ± 0.08
BUN (mmol/L)	10.09 ± 0.19	10.13 ± 0.31	10.01 ± 0.46	10.32 ± 0.36	10.16 ± 0.21	10.11 ± 0.31
CRE (μmol/L)	44.03 ± 1.91	45.21 ± 2.31	42.91 ± 3.33	44.19 ± 2.98	42.98 ± 1.08	43.26 ± 0.09
ALT (IU/L)	38.21 ± 3.41	42.87 ± 2.12	40.23 ± 3.61	41.87 ± 1.97	42.09 ± 2.71	40.98 ± 3.01
AST (IU/L)	129.23 ± 23.12	136.21 ± 11.21	132.98 ± 31.21	138.12 ± 29.12	119.21 ± 38.24	129.12 ± 31.36

* *p* < 0.05 vs. Normal Control.

## Data Availability

Data will be made available on request.

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
