# Peer review of "Yogurt Alleviates Cyclophosphamide-Induced Immunosuppression in Mice through D-Lactate"

_nutrients, 2024, doi:10.3390/nu16091395_

Round 1

Reviewer 1 Report

Comments and Suggestions for Authors

The aim of the study was to determine the effect of commercial yogurt on immunosuppression. To this end female mice were used. The data obtained show that yogurt alleviates cyclophosphamide-induced immunosuppression in mice through D-lactate.

1) Title: is it possible to affirm that the effect of yogurt in the attenuation of immunosuppression is only due to lactate? To my opinion there are only major argument supporting a potential role of lactacte. Consequently the title must be improved.

2) Abstract: must be improved

The different models used must be briefly presented in the abstract. The data obtained must be presented for each model. The conclusion of the abstract must be improved.

3) Introduction: Additionnal refs on the benefits of yogurt on the health are required. Are there differences according to the type of milk considered. Milk contains not only lactate but also fatty acids, phytosterols, vitamins...Nothing is said on these compounds whereas the introduction only focus on lactate. The introduction must be improved.

4) Results

The whole composition of the yogurt must be shown: not only lactate but also other compounds (FAs, phytosterols, Vit, minerals, ...).

Well presented and clear

5) Discussion

Similar criticism than in the discussion. It is focused on lactate and nothing is said on the other compounds of the milk: fatty acids, phytosterols, vitamins

What about the perspectives in terms of functionnal food: which recommandations.

Reviewer 2 Report

Comments and Suggestions for Authors

Dear Authors,

The manuscript entitled: „Yogurt alleviates cyclophosphamide-induced immunosuppression in mice through D-lactate”  aimed to elucidate the immunosuppression-alleviating properties of yogurt and proposed the underlying mechanism was related to the metabolite D-lactate.

The presented manuscript is an example of research with high application potential and significant scientific value. The structure of the reviewed article is well thought out, clear, and in line with the editorial requirements of the journal Nutrients. The introduction provides a good background of the topic. I think the findings of this study are sufficiently described in the context of the published literature. The conclusions are supported by appropriate evidence. It is worth noting that the Authors  used a rich set of research methods and presented the obtained results in the form of extremely legible drawings, which is a big advantage of the analyzed article.

However, I have only one suggestion for Authors. The section : ”Conclusions” needs to be enriched. It should be emphasized that this study is particularly important for the development of the functional food segment, because as one of the few, it describes the impact of such a product on human health.It was a great pleasure to read the text of this manuscript.

From my standpoint, this manuscript is appropriate for publication  in a reputable journal such  as  Nutrients, given the above aspects.  
